# Four weeks *versus* six weeks of ampicillin plus ceftriaxone in *Enterococcus faecalis* native valve endocarditis: A prospective cohort study

**Antonio Ramos-Martínez**[1,2☯]*, **Juan Manuel Pericàs**[3☯], **Ana Fernández-Cruz**[1], **Patricia Muñoz**[4,5], **Maricela Valerio**[4], **Martha Kestler**[4], **Miguel Montejo**[6], **M. Carmen Fariñas**[7], **Dolores Sousa**[8], **Fernando Domínguez**[9], **Guillermo Ojeda-Burgos**[10], **Antonio Plata**[11], **Laura Vidal**[12], **José María Miró**[13]*, **On behalf of the Grupo de Apoyo al Manejo de la Endocarditis Infecciosa en España (GAMES)**[¶]

1 Unidad de Enfermedades Infecciosas, Servicio de Medicina Interna, Hospital Universitario Puerta de Hierro-Majadahonda, Majadahonda, Spain, 2 Instituto de Investigación Sanitaria Puerta de Hierro-Segovia de Arana (IDIPHISA), Madrid, Spain, 3 Dirección Clínica Territorial de Enfermedades Infecciosas, Institut de Recerca Biomèdica de Lleida Fundació Dr. Pifarré, Lleida, Spain, 4 Servicio de Microbiología Clínica y Enfermedades Infecciosas, Hospital General Universitario Gregorio Marañón, Madrid, Spain, 5 Instituto de Investigación Sanitaria Gregorio Marañón, CIBER Enfermedades Respiratorias-CIBERES (CB06/06/0058), Facultad de Medicina, Universidad Complutense de Madrid, Madrid, Spain, 6 Unidad de Enfermedades Infecciosas, Hospital Universitario Cruces, Bilbao, Spain, 7 Diseases Unit, Hospital Universitario Marqués de Valdecilla, University of Cantabria, Santander, Spain, 8 Servicio de Enfermedades Infecciosas, Complejo Hospitalario A Coruña, A Coruña, Spain, 9 Servicio de Cardiología, Hospital Universitario Puerta de Hierro-Majadahonda, Madrid, Spain, 10 Unidad de Gestión Clínica de Enfermedades Infecciosas, Hospital Universitario Virgen de la Victoria, Málaga, Spain, 11 Servicio de Enfermedades Infecciosas, Hospital Regional de Málaga, Málaga, Spain, 12 Servicio de Cardiología, Hospital Universitario Son Espases, Palma de Mallorca, Spain, 13 Infectious Diseases Service, Hospital Clinic-IDIBAPS, University of Barcelona, Barcelona, Spain

☯ These authors contributed equally to this work.
¶ On behalf of the Grupo de Apoyo al Manejo de la Endocarditis Infecciosa en España (GAMES) (see Acknowledgments).
* aramos220@gmail.com (ARM); jmmiro@ub.edu (JMM)

**Data Availability Statement:** The reason for the legal restrictions is related to personal data protection. In UE they are regulated under the

## Abstract

*Enterococcus faecalis* infective endocarditis (EFIE) is a severe disease of increasing incidence. The objective was to analyze whether the outcome of patients with native valve EFIE (NVEFIE) treated with a short course of ampicillin plus ceftriaxone (4wAC) was similar to patients treated according to international guidelines (6wAC). Between January 2008 and June 2018, 1,978 consecutive patients with definite native valve IE were prospectively included in a national registry. Outcomes of patients with NVEFIE treated with 4wAC were compared to those of patients who received 6wAC. Three hundred and twenty-two patients (16.3%) had NVEFIE. One hundred and eighty-three (56.8%) received AC. Thirty-nine patients (21.3%) were treated with 4wAC for four weeks and 70 patients (38.3%) with 6wAC. There were no differences in age or comorbidity. Patients treated 6wAC presented a longer duration of symptoms before diagnosis (21 days, IQR 7–60 days vs. 7 days, IQR 1–22 days; p = 0.002). Six patients presented perivalvular abscess and all of these received 6wAC. Surgery was performed on 14 patients (35.9%) 4wAC and 34 patients (48.6%) 6wAC (p = 0.201). In-hospital mortality, one-year mortality and relapses among 4wAC and 6wAC patients were 10.3% vs. 11.4% (p = 0.851); 17.9% vs. 21.4% (p = 0.682) and 5.1%

resolution 2016/679 of the European Parliament and of the European Council of 27 April 2016 on data protection, whereby personal data cannot be transferred to entities outside the European Union. Patient information is fully available on request through our data manager Ivan Adan (games08@gmail.com).

**Funding:** This work was supported in part by the "Fondo de Investigaciones Sanitarias" (FIS) grant 17/01251 from the "Instituto de Salud Carlos III", Madrid, Spain awarded to JMM. JMM received a personal 80:20 research grant from the Institut d'Investigacions Biomèdiques August Pi i Sunyer (IDIBAPS), Barcelona, Spain, during 2017–19. JMP was member of the Endocarditis Team of the Hospital Clinic of Barcelona, Spain when this project was approved by the GAMES Steering Committee. The funders had no role in study design, data collection and analysis, decision to publish, or preparation of the manuscript.

**Competing interests:** Dr. Ramos-Martínez declares personal fees from Merck, Astellas and Pfizer, outside the submitted work. Dr. Ojeda-Burgos reports personal fees from Merck, personal fees from Janssen Cilag, personal fees from Gilead, outside the submitted work. Dr. Miro reports grants and personal fees from AbbVie, Bristol-Myers Squibb, Contrafect, Genentech, Jansen, Medtronic, Merck, Novartis, Gilead Sciences, and ViiV Healthcare, outside the submitted work. Dr. Juan M Pericas. Dr. Ana Fernández-Cruz, Dr. Patricia Muñoz, Dr. Maricela Valerio, Dr. Martha Kestler, Dr. Miguel Montejo, Dr. Mª Carmen Fariñas, Dr. Dolores Sousa, Dr. Fernando Domínguez, Dr. Antonio Plata and Dr. Laura Vidal have nothing to disclouse. This does not alter our adherence to PLOS ONE policies on sharing data and materials.

**Abbreviations:** TEE, Transesophageal echocardiography; IQR, Interquartile range.

vs. 4.3% (p = 0.833), respectively. In conclusion, a four-week course of AC may be considered as an alternative regimen in NVEFIE, notably in patients with shorter duration of symptoms and those without perivalvular abscess. These results support the performance of a randomized clinical trial to evaluate the efficacy of this short regimen.

## Introduction

*Enterococcus faecalis* is the third most common cause of infective endocarditis (IE) [1]. *E. faecalis* infective endocarditis (EFIE) is characterized by its increasing incidence and tendency to affect elderly patients and those with multiple comorbidities [2, 3].

*E. faecalis* is remarkably resistant to antibiotic-induced eradication, which requires treatment with synergistic antibiotic combinations. Currently, the combination of ampicillin plus ceftriaxone (AC) is the antibiotic treatment of choice for EFIE in several settings, and especially so in Spain [4–7]. With respect to treatment duration, international guidelines recommend a 6 week-course of AC for native valve EFIE [8, 9]. However, these same guidelines accept the use of other antibiotic combinations (e.g. a beta-lactam plus gentamicin) for four weeks when the duration of symptoms is shorter than 3 months. Evidence is lacking from studies comparing the efficacy of shorter courses of AC with respect to the recommended duration of 6 weeks for the treatment of EFIE. However, the use of a shortened course of AC is relatively common for treating non-complicated native EFIE.

The aim of the present study was to analyze whether the clinical outcomes of patients suffering from native valve EFIE treated with a short-course regimen of AC (four weeks) were similar to those of patients treated for the recommended six weeks.

## Methods

Between January 2008 and June 2018, 3,830 consecutive patients were prospectively included in the "Spanish Collaboration on Endocarditis—Grupo de Apoyo al Manejo de la Endocarditis Infecciosa en España (GAMES)", a registry maintained by 27 Spanish hospitals. Multidisciplinary teams completed standardized case report forms with IE episode and follow-up data that included clinical, microbiological and echocardiographic sections [10, 11]. The study was approved by regional ethics committee (Comité Ético de Investigación Clínica Regional de la Comunidad de Madrid with approval code: 07/18). Written informed consent was obtained from all patients.

### Definitions

IE was defined according to the modified Duke criteria [12]. Microbiological diagnoses were determined by blood or valve culture and/or molecular techniques [10]. Transthoracic (TTE) and transesophageal echocardiography (TEE) were performed on patients with clinical or microbiological suspicion of IE according to European guidelines, or to diagnose valve dysfunction and intracardiac complications such as abscess, vegetation, pseudoaneurysm and fistula [8, 13, 14]. Hospital-acquired IE was defined as either IE manifesting >48 hours after hospital admission or IE acquired in association with a significant invasive procedure performed within 6 months prior to diagnosis. The EuroScore and LogEuroScore were used to assess surgical risk [15]. All the necessary variables were collected to calculate the Charlson Comorbidity Index [16].

For the purposes of this study, patients with definite native valve EFIE were selected from the GAMES database. Patients who only met possible Duke IE criteria as well as right-sided IE cases without concomitant left-side infection were excluded from the study. Fourteen patients had been included in a previous investigation [17]. Patients classified as treated with AC did not receive gentamicin at any time. The dose of ampicillin was 12 grams per day in 4 or 6 doses (adjusted for renal function) and ceftriaxone was 2 g every 12 hours, both intravenously. The decision about the duration of treatment was taken by the responsible physicians at the beginning of the treatment but could also be taken during the patient's clinical evolution. Patients were classified according to the length of treatment: patients who received 28 ± 4 days of AC antibiotic treatment were included in the short-course group, whereas patients who received 42 ± 6 days of treatment were included in the long- or conventional-treatment group The date of the first negative blood culture was considered in calculating the duration of treatment. Patients with a length of treatment that did not fit any of these definitions were excluded.

In-hospital mortality was defined as death, irrespective of cause, that occurred during hospital admission in patients who had completed antibiotic treatment. Patients who died before completing their allotted course of treatment were excluded from the analysis. The Cockcroft-Gault equation was used to calculate creatinine clearance [18]. Pre-episode renal insufficiency was defined as plasma creatinine over 1.4 mg/dl. New or worsening renal insufficiency during the IE episode was defined as an increase of baseline creatinine clearance or plasma creatinine by at least 25%, or creatinine levels over 1.4 mg/dl when a previous analysis had been normal. Persistent bacteremia was defined as positive blood cultures more than seven days after effective antibiotic therapy. Endocarditis relapse was defined a new episode of EFIE during the first year after finishing treatment.

Data from patients with native valve EFIE were analyzed, including clinical manifestations at IE presentation, pathogens, therapy, morbidity and mortality during hospitalization and during the first year after hospitalization. Follow-up information was obtained via programmed medical follow-up, by telephone or through written or electronic correspondence with the patients or their primary-care physician.

## Statistical analysis

Quantitative variables were reported as median and interquartile range (IQR), qualitative variables were reported as figures and percentages. Continuous variables were compared using the Mann- Whitney test, because they had a non-normal distribution. Categorical variables were summarized as percentages and continuous variables as medians and interquartile ranges. Categorical variables were compared using the chi-square test (or Fisher's exact test where necessary). Survival analysis was performed using Kaplan-Meier, and survival curves (showing the survival of patients from diagnosis until the end of the first year) were compared using the log-rank test. Given the limited number of patients in the study, no multivariate logistic regression analyses were performed. A 2-sided $P<0.05$ was considered to be statistically significant. All statistical analyses were performed using SPSS software version 18 (SPSS Inc., Chicago, Illinois, USA).

## Results

Three hundred and twenty-two patients (16.3%) had definite native valve EFIE. Most patients received AC (183 patients, 56.8%), while ampicillin plus gentamicin (AG) was administered to 73 patients (22.7%), and 66 patients (20.5%) were treated with other options. The patient flowchart is shown in Fig 1.

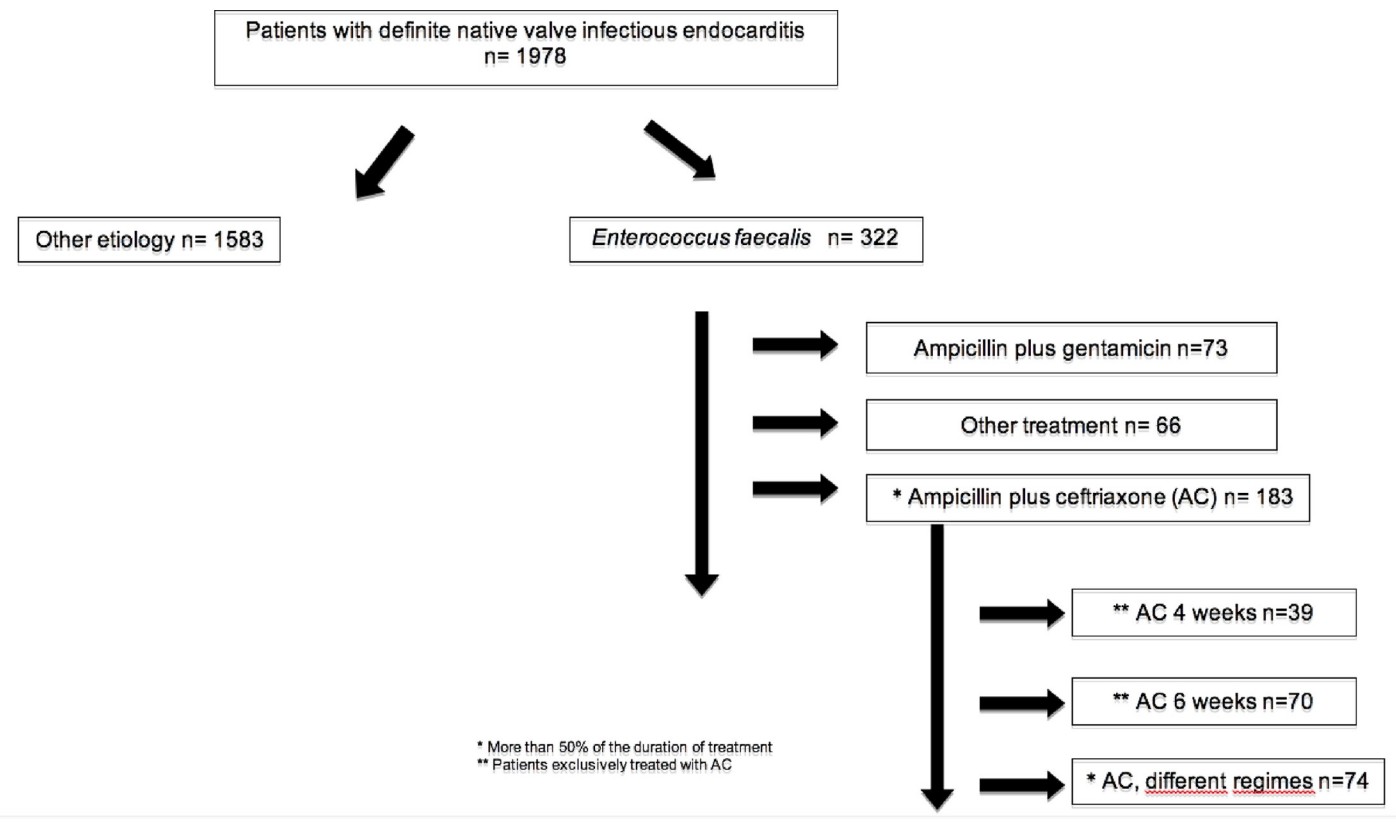

**Fig 1. Chart showing patients presenting with definite native valve infective endocarditis according to etiology and treatment (GAMES cohort).**

Among the subset of 183 patients treated with AC, 39 patients were treated exclusively with these antibiotics for four weeks and 70 patients for six weeks. The differences between the short treatment (four weeks) and conventional treatment group (six weeks) are shown in Tables 1

**Table 1. Baseline characteristics of patients with native valve E. faecalis infective endocarditis (EFIE) treated with ampicillin plus ceftriaxone according to the duration of antibiotic treatment.**

|  | 4 weeks (n = 39) | 6 weeks (N = 70) | P |
|---|---|---|---|
| Age (years) | 75 (67–80) | 72 (63–78) | 0.222 |
| Male gender | 26 (66.7) | 50 (71.4) | 0.604 |
| Hospital-acquired | 10 (25.6) | 22 (31.4) | 0.676 |
| Non-nosocomial health-care related | 4 (10.2) | 10 (14.2) | 0.761 |
| Diabetes mellitus | 11 (28.2) | 20 (28.6) | 0.404 |
| Coronary disease | 8 (20.5) | 22 (31.4) | 0.284 |
| Peripheral arterial disease | 3 (7.6) | 8 (11.4) | 0.775 |
| Cerebrovascular disease | 6 (15.4) | 5 (7.1) | 0.15 |
| Previous renal failure | 11 (28.2) | 17 (24.3) | 0.653 |
| Chronic liver disease | 4 (10.3) | 7 (10.0) | 0.931 |
| Injecting drug user | 0 | 1 (1.4) | 0.648 |
| Neoplasia | 9 (23.7) | 14 (20.0) | 0.655 |
| Age-adjusted Charlson Comorbidity Index (points) | 6 (4–7) | 6 (4–8) | 0.716 |

Quantitative variables are reported with median and interquartile range.

**Table 2. Clinical presentation an outcome of patients with native valve E. faecalis infective endocarditis (EFIE) treated with ampicillin plus ceftriaxone according to the duration of antibiotic treatment.**

| | 4 weeks (n = 39) | 6 weeks (N = 70) | P |
|---|---|---|---|
| Site of infection | | | |
| Mitral | 22 (56.4) | 48 (68.6) | 0.204 |
| Aortic | 17 (43.6) | 33 (47.1) | 0.721 |
| Tricuspid | 1 (2.6) | 5 (7.1) | 0.417 |
| Pulmonary | 0 | 0 | - |
| Duration of symptoms before diagnosis (days) | 7 (1–22) | 21 (7–60) | 0.002 |
| Septic shock | 2 (5.1) | 4 (5.7) | 0.922 |
| Persistent bacteremia | 6 (15.4) | 10 (14.7) | 0.925 |
| CNS vascular events | 5 (12.8) | 12 (17.1) | 0.551 |
| Embolism | 8 (20.5) | 16 (22.9) | 0.717 |
| Heart failure | 18 (46.2) | 34 (48.6) | 0.809 |
| New or worsening renal insufficiency | 10 (25.6) | 20 (28.6) | 0.743 |
| Echocardiographic findings | | | |
| Vegetation | 34 (87.1) | 62 (88.6) | 0.925 |
| Median size vegetation (IQR) | 12 (8–18) | 14 (7–20) | 0.471 |
| Perivalvular abscess | 0 | 6 (8.6) | 0.060 |
| Valve perforation or rupture | 4 (10.3) | 17 (24.3) | 0.060 |
| Pseudoaneurysm | 1 (2.6) | 3 (4.3) | 0.549 |
| Intracardiac fistula | 0 | 0 | - |
| Surgical indication | 17 (43.6) | 45 (64.3) | 0.085 |
| Surgery performed | 14 (35.9) | 34 (48.6) | 0.201 |
| Duration of antibiotic treatment | 32 (28–42) | 42 (42–45) | <0.01 |
| Treatment with ampicillin (days) | 29 (28–32) | 42 (41–45) | <0.01 |
| Treatment with ceftriaxone (days) | 29 (28–30) | 43 (41–45) | <0.01 |
| Hospital stay (days) | 40 (27–54) | 51 (44–66) | 0.001 |
| EFIE relapse[1] | 2 (5.1) | 3 (4.3) | 0.833 |
| In-hospital mortality | 4 (10.3) | 8 (11.4) | 0.851 |
| One-year mortality | 7 (17.9) | 15 (21.4) | 0.682 |

CNS: central nervous system. Quantitative variables are reported with median and interquartile range.

[1]See text for relapses.

and 2. In terms of baseline conditions, age and comorbidity were similar in patients treated for four weeks and in those treated for six weeks. Chronic liver disease was present in 10.3% (n = 4) of patients receiving short treatment and 10% (n = 7) of patients receiving long treatment. Six patients presented perivalvular abscess and all of these were treated for six weeks (P = 0.072, Table 2). With regard to prognosis related to echocardiographic findings, the presence of perivalvular abscess, valve perforation or a vegetation size greater than 10 mm was associated with an increased risk of recurrence and increased mortality during admission or at one year. Surgery was performed on 14 patients (35.9%) receiving short treatment and on 34 patients (48.6%) receiving long treatment (P = 0.201).

Outcomes were similar in the two groups: there were no differences regarding the incidence of persistent bacteremia, embolisms, relapses, in-hospital mortality or mortality during the first year (Table 2, Fig 2). There were five relapses, two in 4wAC group (both during the first six months after treatment) and three in the 6wAC group (both during the first six months after treatment). Two patients had a history of liver cirrhosis (one in each group) and one of

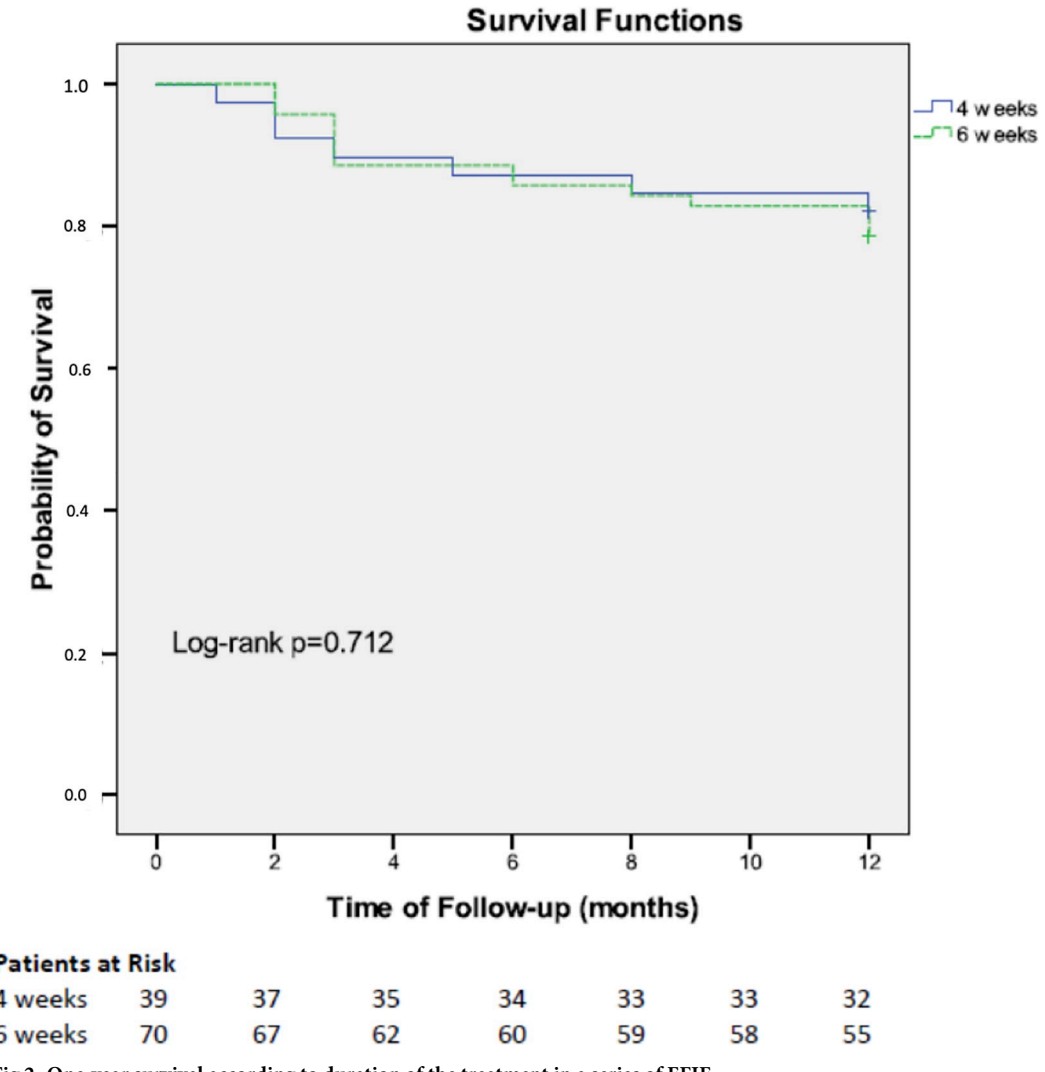

**Fig 2. One-year survival according to duration of the treatment in a series of EFIE.**

them, who suffered two relapses, has already been communicated in a previous publication [17]. None of the cases had paravalvular abscesses.

Adverse effects attributable to ceftriaxone were leukopenia (two patients, both 6wAC group), glomerulonephritis (one patient, 6wAC group), diarrhea associated with antibiotics without *Clostridiodes difficile* infection (one patient, 4wAC group). Of the 74 patients who received CA for more than 50% of the time but did not meet the criteria for inclusion in the 4- or 6-week treatment groups, 29 patients received at least 4 weeks of treatment. In these patients, mortality during admission was 5.5% (2 patients), mortality during the first year was 22.2% (8 patients) and no cases of recurrence were recorded

## Discussion

This is the first cohort study to primarily focus on the prognosis of patients with EFIE treated with a short course of AC. It shows that the prognosis of patients with native valve EFIE treated for 4 weeks with AC is similar to those treated for 6 weeks. However, some differences in the

clinical characteristics of patients between the 4-week and 6-week groups may limit the applicability of this result to selected cases.

Both AG and AC have been shown to be synergistic combinations *in vitro* and effective antibiotic treatments *in vivo* for EFIE [1, 4, 5, 19, 20]. However, guidelines recommend different durations for each of these combinations when treating native valve EFIE: four or six weeks for AG and six weeks for AC [8, 9]. The median duration of treatment with AC used in the first two studies to compare the efficacy of AC versus AG was six weeks [4, 21]. As a result, international guidelines incorporated the 6-week protocol as the universal duration for courses of AC to treat both native and prosthetic valve EFIE [8, 9]. The study by Fernandez-Hidalgo et al confirmed that AC given for a median of six weeks was an effective therapy for EFIE and safer than four to six weeks of ampicillin plus gentamicin. However, this study did not clarify whether four weeks of ampicillin plus ceftriaxone would be effective against uncomplicated native valve EFIE [4]. A previous single-center study compared 4-week with 6-week treatments for EFIE by analyzing patients treated with AC and AG according to the duration of antibiotic treatment [17]. Neither that study nor ours detected higher mortality (either during hospital admission or the first year) associated with short-course treatments. However, in the aforementioned study, relapses were more frequent with short-course treatments (with either AC or AG) when analyzing both native and prosthetic valves. Interestingly, relapses were no higher when analyzing only patients with native valve EFIE who were treated for four weeks. It is also worth noting that the relapse rate in patients who received treatment for four weeks in our study (5%) was similar to the other series of EFIE based on a treatment with AC lasting at least six weeks (3–8%) [1, 4, 5, 21].

The propensity of patients with chronic liver disease to develop enterococcal invasive and relapsing infections should be taken into consideration when deciding the appropriate length of EFIE treatment [17, 22]. Among the patients included in this study, only 10% had chronic liver disease, which limits significant conclusions in this regard. Two of the patients who relapsed had chronic liver disease. In any case, considering that these patients have a high surgical risk, it is of paramount importance to try to minimize the risk of relapse, since this could leave surgery as the patient's only recourse. Consequently, we have to express our doubts that a four-week regimen would be recommendable for cirrhotic patients.

Seminal studies that provided evidence for establishing the length of EFIE treatment with beta-lactams plus aminoglycosides showed that longer periods of symptoms have generally been associated to an increased risk of relapse and death [23, 24]. It was suggested that longer illness is due to enterococci that grow relatively slowly and produce dense vegetations which are less accessible to antimicrobial agents [23]. Despite the fact that none of studied patients presented symptoms for more than 3 months prior to diagnosis, the duration of symptoms was longer in those who received six weeks compared to those who received four weeks of treatment. It is possible that long treatments were selected for patients with longer duration of symptoms due to its relationship with poor prognoses [23]. Therefore, our results may not substantiate the use of short courses in patients with longer duration of symptoms.

All patients with perivalvular abscess received six weeks of AC, which is an expected result. It would be seen as unsafe to select a short course of treatment of either AG or AC in these patients who also had indications for surgical treatment [25]. In our series, patients with valve perforation or rupture and those who underwent surgery also tended to receive longer treatment. These clinical differences between the two groups could preclude recommending a short course of AC in patients with some of the above-mentioned characteristics.

We found a significant shortening of the hospital stay in short course treated patients. The saving in patient complications and hospital stay on account of short-course treatment has already been a common goal of various IE studies. [26, 27]. The potential protective effect of

shorter courses of beta-lactams, ranging from lower rates of antibiotic-related toxicity to the risk of suffering from *C. difficile* infections or for the selection of vancomycin-resistant *E. faecium* [28–30].

## Limitations

Some limitations of the study should be noted. The main limitation is that it was not a randomized clinical trial, which could have determined that the probability of receiving a short or long treatment depended on certain patient characteristics. Analysis based on "intention to treat" could not be done because this variable was not collected in the study. The duration of treatment actually completed by each patient was primarily considered. In addition, most of the institutions participating in the GAMES registry are tertiary university hospitals that receive a substantial number of patients from other centers (most of which do not have facilities for cardiac surgery), which could represent a certain selection bias. And finally, the statistical power of this study is limited due to the low number of patients included. This limitation may have prevented detecting a difference in prognosis in relation to the duration of treatment. These limitations induce caution when drawing conclusions regarding the efficacy of short-course AC treatments. In spite of this limitations, the importance of the pathology studied and the difficulties in carrying out clinical trials in this type of patient allow, in our opinion, to anticipate the possibility of a short treatment for some selected patients.

## Conclusions

In conclusion, due to very similar rates of relapse and mortality in patients with native valve EFIE treated with AC for 4 and 6 weeks, this study suggests that a short course of AC might be sufficient to treat native valve EFIE. Due to clinical differences between the two groups studied, a short course of antibiotics cannot be recommended for patients presenting with longer duration of symptoms or perivalvular abscess. Further research is required to validate these results as well as to elucidate whether short-course treatments should be recommended to treat all native valve EFIE, or only to treat selected patients with specific features that present a more favorable prognosis which is still to be defined. This study could be the rationale for performing a randomized clinical trial in non-complicated native valve EFIE.

## Acknowledgments

**Members of GAMES: Hospital Costa del Sol**, (Marbella): Fernando Fernández Sánchez, Mariam Noureddine, Gabriel Rosas, Javier de la Torre Lima; **Hospital Universitario de Cruces**, (Bilbao): Roberto Blanco, María Victoria Boado, Marta Campaña Lázaro, Alejandro Crespo, Josune Goikoetxea, José Ramón Iruretagoyena, Josu Irurzun Zuazabal, Leire López-Soria, Miguel Montejo, Javier Nieto, David Rodrigo, Regino Rodríguez, Yolanda Vitoria, Roberto Voces; **Hospital Universitario Virgen de la Victoria**, (Málaga): Mª Victoria García López, Radka Ivanova Georgieva, Guillermo Ojeda, Isabel Rodríguez Bailón, Josefa Ruiz Morales; **Hospital Universitario Donostia-Policlínica Gipuzkoa**, (San Sebastián): Ana María Cuende, Tomás Echeverría, Ana Fuerte, Eduardo Gaminde, Miguel Ángel Goenaga, Pedro Idígoras, José Antonio Iribarren, Alberto Izaguirre Yarza, Xabier Kortajarena Urkola, Carlos Reviejo; **Hospital General Universitario de Alicante**, (Alicante): Rafael Carrasco, Vicente Climent, Patricio Llamas, Esperanza Merino, Joaquín Plazas, Sergio Reus; **Complejo Hospitalario Universitario A Coruña**, (A Coruña): Nemesio Álvarez, José María Bravo-Ferrer, Laura Castelo, José Cuenca, Pedro Llinares, Enrique Miguez Rey, María Rodríguez Mayo, Efrén Sánchez, Dolores Sousa Regueiro; **Complejo Hospitalario Universitario de Huelva**, (Huelva): Francisco Javier Martínez; **Hospital Universitario de Canarias**, (Canarias): Mª del Mar Alonso,

Beatriz Castro, Teresa Delgado Melian, Javier Fernández Sarabia, Dácil García Rosado, Julia González González, Juan Lacalzada, Lissete Lorenzo de la Peña, Alina Pérez Ramírez, Pablo Prada Arrondo, Fermín Rodríguez Moreno; **Hospital Regional Universitario de Málaga**, (Málaga): Antonio Plata Ciezar, José Mª Reguera Iglesias; **Hospital Universitario Central Asturias**, (Oviedo): Víctor Asensi Álvarez, Carlos Costas, Jesús de la Hera, Jonnathan Fernández Suárez, Lisardo Iglesias Fraile, Víctor León Arguero, José López Menéndez, Pilar Mencia Bajo, Carlos Morales, Alfonso Moreno Torrico, Carmen Palomo, Begoña Paya Martínez, Ángeles Rodríguez Esteban, Raquel Rodríguez García, Mauricio Telenti Asensio; **Hospital Clínic-IDIBAPS, Universidad de Barcelona**, (Barcelona): Manuel Almela, Juan Ambrosioni, Manuel Azqueta, Mercè Brunet, Marta Bodro, Ramón Cartañá, Carlos Falces, Guillermina Fita, David Fuster, Cristina García de la Mària, Delia García-Pares, Marta Hernández-Meneses, Jaume Llopis Pérez, Francesc Marco, José M. Miró, Asunción Moreno, David Nicolás, Salvador Ninot, Eduardo Quintana, Carlos Paré, Daniel Pereda, Juan M. Pericás, José L. Pomar, José Ramírez, Irene Rovira, Elena Sandoval, Marta Sitges, Dolors Soy, Adrián Téllez, José M. Tolosana, Bárbara Vidal, Jordi Vila; **Hospital General Universitario Gregorio Marañón**, (Madrid): Iván Adán, Javier Bermejo, Emilio Bouza, Daniel Celemín, Gregorio Cuerpo Caballero, Antonia Delgado Montero, Ana García Mansilla, Mª Eugenia García Leoni, Víctor González Ramallo, Martha Kestler Hernández, Amaia Mari Hualde, Mercedes Marín, Manuel Martínez-Sellés, Patricia Muñoz, Cristina Rincón, Hugo Rodríguez-Abella, Marta Rodríguez-Créixems, Blanca Pinilla, Ángel Pinto, Maricela Valerio, Pilar Vázquez, Eduardo Verde Moreno; **Hospital Universitario La Paz**, (Madrid): Isabel Antorrena, Belén Loeches, Alejandro Martín Quirós, Mar Moreno, Ulises Ramírez, Verónica Rial Bastón, María Romero, Araceli Saldaña; **Hospital Universitario Marqués de Valdecilla**, (Santander): Jesús Agüero Balbín, Carlos Armiñanzas Castillo, Ana Arnaiz, Francisco Arnaiz de las Revillas, Manuel Cobo Belaustegui, María Carmen Fariñas, Concepción Fariñas-Álvarez, Rubén Gómez Izquierdo, Iván García, Claudia González Rico, Manuel Gutiérrez-Cuadra, José Gutiérrez Díez, Marcos Pajarón, José Antonio Parra, Ramón Teira, Jesús Zarauza; **Hospital Universitario Puerta de Hierro**, (Madrid): Jorge Calderón Parra, Marta Cobo, Fernando Domínguez, Alberto Fortaleza, Pablo García Pavía, Jesús González, Ana Fernández Cruz, Elena Múñez, Antonio Ramos, Isabel Sánchez Romero; **Hospital Universitario Ramón y Cajal**, (Madrid): Tomasa Centella, José Manuel Hermida, José Luis Moya, Pilar Martín-Dávila, Enrique Navas, Enrique Oliva, Alejandro del Río, Jorge Rodríguez-Roda Stuart, Soledad Ruiz; **Hospital Universitario Virgen de las Nieves**, (Granada): Carmen Hidalgo Tenorio; **Hospital Universitario Virgen Macarena**, (Sevilla): Manuel Almendro Delia, Omar Araji, José Miguel Barquero, Román Calvo Jambrina, Marina de Cueto, Juan Gálvez Acebal, Irene Méndez, Isabel Morales, Luis Eduardo López-Cortés; **Hospital Universitario Virgen del Rocío**, (Sevilla): Arístides de Alarcón, Emilio García, Juan Luis Haro, José Antonio Lepe, Francisco López, Rafael Luque; **Hospital San Pedro**, (Logroño): Luis Javier Alonso, Pedro Azcárate, José Manuel Azcona Gutiérrez, José Ramón Blanco, Antonio Cabrera Villegas, Lara García-Álvarez, José Antonio Oteo, Mercedes Sanz; **Hospital de la Santa Creu i Sant Pau**, (Barcelona): Natividad de Benito, Mercé Gurguí, Cristina Pacho, Roser Pericas, Guillem Pons; **Complejo Hospitalario Universitario de Santiago de Compostela**, (A Coruña): M. Álvarez, A. L. Fernández, Amparo Martínez, A. Prieto, Benito Regueiro, E. Tijeira, Marino Vega; **Hospital Santiago Apóstol**, (Vitoria): Andrés Canut Blasco, José Cordo Mollar, Juan Carlos Gainzarain Arana, Oscar García Uriarte, Alejandro Martín López, Zuriñe Ortiz de Zárate, José Antonio Urturi Matos; **Hospital SAS Línea de la Concepción**, (Cádiz): García-Domínguez Gloria, Sánchez-Porto Antonio; **Hospital Clínico Universitario Virgen de la Arrixaca** (Murcia): José Mª Arribas Leal, Elisa García Vázquez, Alicia Hernández Torres, Ana Blázquez, Gonzalo de la Morena Valenzuela; **Hospital de Txagorritxu**, (Vitoria): Ángel Alonso, Javier Aramburu, Felicitas

Elena Calvo, Anai Moreno Rodríguez, Paola Tarabini-Castellani; **Hospital Virgen de la Salud,** (Toledo): Eva Heredero Gálvez, Carolina Maicas Bellido, José Largo Pau, Mª Antonia Sepúlveda, Pilar Toledano Sierra, Sadaf Zafar Iqbal-Mirza; **Hospital Rafael Méndez,** (Lorca-Murcia):, Eva Cascales Alcolea, Ivan Keituqwa Yañez, Julián Navarro Martínez, Ana Peláez Ballesta; **Hospital Universitario San Cecilio** (Granada): Eduardo Moreno Escobar, Alejandro Peña Monje, Valme Sánchez Cabrera, David Vinuesa García; **Hospital Son Llátzer** (Palma de Mallorca): María Arrizabalaga Asenjo, Carmen Cifuentes Luna, Juana Núñez Morcillo, Mª Cruz Pérez Seco, Aroa Villoslada Gelabert; **Hospital Universitario Miguel Servet** (Zaragoza): Carmen Aured Guallar, Nuria Fernández Abad, Pilar García Mangas, Marta Matamala Adell, Mª Pilar Palacián Ruiz, Juan Carlos Porres; **Hospital General Universitario Santa Lucía** (Cartagena): Begoña Alcaraz Vidal, Nazaret Cobos Trigueros, María Jesús Del Amor Espín, José Antonio Giner Caro, Roberto Jiménez Sánchez, Amaya Jimeno Almazán, Alejandro Ortín Freire, Monserrat Viqueira González; **Hospital Universitario Son Espases** (Palma de Mallorca): Pere Pericás Ramis, Mª Ángels Ribas Blanco, Enrique Ruiz de Gopegui Bordes, Laura Vidal Bonet; **Complejo Hospitalario Universitario de Albacete** (Albacete): Mª Carmen Bellón Munera, Elena Escribano Garaizabal, Antonia Tercero Martínez, Juan Carlos Segura Luque; **Hospital Universitario Terrassa:** Cristina Badía, Lucía Boix Palop, Mariona Xercavins, Sónia Ibars. **Hospital Universitario Dr. Negrín** (Gran Canaria): Eloy Gómez Nebreda, Ibalia Horcajada Herrera, Irene Menduiña Gallego. **Complejo Hospitalario Universitario Insular Materno Infantil** (Las Palmas de Gran Canaria): Héctor Marrero Santiago, Isabel de Miguel Martínez, Elena Pisos Álamo. **Hospital Universitario 12 de Octubre** (Madrid): Carmen Díaz Pedroche, Fernando Chaves, Santiago de Cossío, Francisco López Medrano, Mª Jesús López, Javier Solera, Jorge Solís. **Hospital Universitari Bellvitge** (Barcelona): Carmen Ardanuy, Guillermo Cuervo Requena, Sara Grillo, Alejandro Ruiz Majoral.

**Lead author of the group: Antonio Ramos-Martínez.** aramos220@gmail.com. [1] Unidad de Enfermedades Infecciosas. Servicio de Medicina Interna. Hospital Universitario Puerta de Hierro-Majadahonda. [2] Instituto de Investigación Sanitaria Puerta de Hierro-Segovia de Arana (IDIPHISA). Joaquín Rodrígo 2, 28220. Madrid, Spain.

## Author Contributions

**Conceptualization:** Antonio Ramos-Martínez, Juan Manuel Pericàs, Ana Fernández-Cruz, Patricia Muñoz, M. Carmen Fariñas, Fernando Domínguez, Antonio Plata, José María Miró.

**Data curation:** Antonio Ramos-Martínez, Juan Manuel Pericàs, Ana Fernández-Cruz, Patricia Muñoz, Maricela Valerio, Martha Kestler, Miguel Montejo, M. Carmen Fariñas, Dolores Sousa, Fernando Domínguez, Guillermo Ojeda-Burgos, Antonio Plata, Laura Vidal, José María Miró.

**Formal analysis:** Antonio Ramos-Martínez, Juan Manuel Pericàs, Ana Fernández-Cruz, Patricia Muñoz, Maricela Valerio, Martha Kestler, Miguel Montejo, M. Carmen Fariñas, Dolores Sousa, José María Miró.

**Funding acquisition:** Antonio Ramos-Martínez, Juan Manuel Pericàs.

**Investigation:** Antonio Ramos-Martínez, Juan Manuel Pericàs, Maricela Valerio, José María Miró.

**Methodology:** Antonio Ramos-Martínez, Juan Manuel Pericàs, Ana Fernández-Cruz, Maricela Valerio, José María Miró.

**Project administration:** Antonio Ramos-Martínez, Juan Manuel Pericàs.

**Resources:** Antonio Ramos-Martínez, Juan Manuel Pericàs.

**Software:** Antonio Ramos-Martínez.

**Supervision:** Antonio Ramos-Martínez, Juan Manuel Pericàs, Martha Kestler, Laura Vidal, José María Miró.

**Validation:** Antonio Ramos-Martínez, Juan Manuel Pericàs.

**Visualization:** Antonio Ramos-Martínez, Juan Manuel Pericàs, Guillermo Ojeda-Burgos, José María Miró.

**Writing – original draft:** Antonio Ramos-Martínez, Juan Manuel Pericàs, Fernando Domínguez, José María Miró.

**Writing – review & editing:** Antonio Ramos-Martínez, Juan Manuel Pericàs, Ana Fernández-Cruz, Patricia Muñoz, Maricela Valerio, Martha Kestler, Miguel Montejo, M. Carmen Fariñas, Dolores Sousa, Fernando Domínguez, Guillermo Ojeda-Burgos, Antonio Plata, Laura Vidal, José María Miró.

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
