## [Decision Letter · Decision Letter 0]

26 May 2020

PONE-D-20-05909

Four weeks versus six weeks of ampicillin plus ceftriaxone in Enterococcus faecalis native valve endocarditis: A prospective cohort study.

PLOS ONE

Dear Dr. Ramos,

Thank you for submitting your manuscript to PLOS ONE. After careful consideration, we feel that it has merit but does not fully meet PLOS ONE’s publication criteria as it currently stands. Therefore, we invite you to submit a revised version of the manuscript that addresses the points raised during the review process.

We look forward to receiving your revised manuscript.

Kind regards,

Cécile Oury

Academic Editor

PLOS ONE

Journal Requirements:

2. Please provide the full names of the Institutional Review Boards s that approved the study in your manuscript.

3. Please specify in your ethics statement whether participant consent was written or verbal. If verbal, please also specify: 1) whether the ethics committee approved the verbal consent procedure, 2) why written consent could not be obtained, and 3) how verbal consent was recorded.” Do not ping with follow up unless there are questions

'This work was supported in part by the “Fondo de Investigaciones Sanitarias” (FIS)

grant 17/01251 from the “Instituto de Salud Carlos III”, Madrid, Spain awarded to

JMM. JMM received a personal 80:20 research grant from the Institut d’Investigacions

Biomèdiques August Pi i Sunyer (IDIBAPS), Barcelona, Spain, during 2017–19. JMP was

member of the Endocarditis Team of the Hospital Clinic of Barcelona, Spain when this

project was approved by the GAMES Steering Committee.'

'The funders had no role in study design, data collection and analysis, decision to

publish, or preparation of the manuscript.'

'Dr. Ramos-Martínez declares personal fees from Merck, Astellas and Pfizer, outside

the submitted work. Dr. Ojeda-Burgos reports personal fees from Merck, personal fees

from Janssen Cilag, personal fees from Gilead, outside the submitted work. Dr. Miro

reports grants and personal fees from AbbVie, Bristol-Myers Squibb, Contrafect,

Genentech, Jansen, Medtronic, Merck, Novartis, Gilead Sciences, and ViiV Healthcare,

outside the submitted work. Dr. Juan M Pericas. Dr. Ana Fernández-Cruz, Dr. Patricia

Muñoz, Dr. Maricela Valerio, Dr. Martha Kestler, Dr. Miguel Montejo, Dr. Mª Carmen

Fariñas, Dr. Dolores Sousa, Dr. Fernando Domínguez, Dr. Antonio Plata and Dr. Laura

Vidal have nothing to disclouse'

6. Please amend the manuscript submission data (via Edit Submission) to include author Guillermo Ojeda-Burgos MD PhD

Additional Editor Comments (if provided):

Reviewers' comments:

Reviewer's Responses to Questions

**Comments to the Author**

1. Is the manuscript technically sound, and do the data support the conclusions?

Reviewer #1: Yes

2. Has the statistical analysis been performed appropriately and rigorously? 

Reviewer #1: No

3. Have the authors made all data underlying the findings in their manuscript fully available?

Reviewer #1: Yes

4. Is the manuscript presented in an intelligible fashion and written in standard English?

Reviewer #1: Yes

5. Review Comments to the Author

Reviewer #1: Dear authors,

Your work addresses an important clinical issue with regard to treating patients with NVEFIE. Interesting work.

Major issues:

1. Statistical issue: p8, 202-204. It is stated that continous varibales will be reported as a median with IQR and that a student's t-test will be used to compare these values. If you report values as a median with IQR, it means that your data was not normally distributed. Did you check if the continous variables had a normal distribution? In that case, it is not correct to use a student's t-test and a non-parametric test should be used.

2. Statistical issue: p8, 204. The aim of this study is to test whether the clinical outcome of patients that received AC for 4 weeks is similar to those that received AC for 6 weeks. Student t-tests are made to test if there is a significant difference, not to test equivalance. Did you perform an equivalence test or can this be conducted?

3. p3, 100-101. In the abstract you mention that patients that received AC for 4 weeks had a longer duration of symptoms (21 days) than patients thet received AC for 6 weeks (7 days). In table 1, the values are the opposite. Is this a mistake? Should be corrected, because it is concluded that treatment with AC for 4 weeks may be considered in patients with a shorter duration of symptoms.

Minor issues:

4. Does severity of symptoms correlate with a worse outcome? For example, perivalvular abscesses, vegetation size, valve perforation, ...

5. Data represented in table 1 will be more clear if this table is divided in to two tables: one with the baseline characteristics and another one with the data on outcome.

6. PLOS authors have the option to publish the peer review history of their article (what does this mean?). If published, this will include your full peer review and any attached files.

Reviewer #1: No

---

## [Author Response · Author response to Decision Letter 0]

30 Jun 2020

Dr. Antonio Ramos

Infectious Diseases Unit (Internal Medicine)

HU Puerta de Hierro

Majadahonda. Madrid. Spain. 28222

14th June 2020

Dear Editor

Thank you for revising our article entitled: "Four weeks versus six weeks of ampicillin plus ceftriaxone in Enterococcus faecalis native valve endocarditis: A prospective cohort study” (PONE-D-20-05909).

Please find below the responses to the suggestions and recommendations made by the reviewer. We hope that the revised version satisfies their concerns. To highlight the changes in the manuscript they have been written in blue. In addition, while revising the manuscript we have also taken the opportunity to correct a few grammatical and typographic errors which in no way affect the scientific content of the article.

Journal Requirements

We have tried to all meet PLOS ONE's requirements. The online system on the manuscript submission has not allowed us to enter the funding information correctly. Please update our funding statement to include: 'This work was supported in part by the “Fondo de Investigaciones Sanitarias” (FIS) grant 17/01251 from the “Instituto de Salud Carlos III”, Madrid, Spain awarded to JMM. JMM received a personal 80:20 research grant from the Institut d’Investigacions Biomèdiques August Pi i Sunyer (IDIBAPS), Barcelona, Spain, during 2017–19. JMP was member of the Endocarditis Team of the Hospital Clinic of Barcelona, Spain when this project was approved by the GAMES Steering Committee.'

All authors declare no potentially competing interests that occurred within 5 years of conducting the research under consideration nor preparing the article for publication.

Reviewer

1. Statistical issue: p8, 202-204. It is stated that continuous variables will be reported as a median with IQR and that a student's t-test will be used to compare these values. If you report values as a median with IQR, it means that your data was not normally distributed. Did you check if the continuous variables had a normal distribution? In that case, it is not correct to use a student's t-test and a non-parametric test should be used.

We appreciate the reviewer's comment. The quantitative variables studied did not present a normal distribution, so it was necessary to use the Mann-Whitney test. Accordingly, the paragraph referring to statistical analysis has been modified. 

2. Statistical issue: p8, 204. The aim of this study is to test whether the clinical outcome of patients that received AC for 4 weeks is similar to those that received AC for 6 weeks. Student t-tests are made to test if there is a significant difference, not to test equivalence. Did you perform an equivalence test or can this be conducted?

In fact, we did not use the Student t-tests to ascertain whether there was a difference between the two groups. What we intended to do was to check whether the prognosis of the patients (assessed by the rate of disease recurrence or mortality) was similar with both therapeutic regimes. We considered that the most appropriate way to check that the two regimes were not different in our patients was to use a qualitative test such as Fisher's.

3. p3, 100-101. In the abstract you mention that patients that received AC for 4 weeks had a longer duration of symptoms (21 days) than patients that received AC for 6 weeks (7 days). In table 1, the values are the opposite. Is this a mistake? Should be corrected, because it is concluded that treatment with AC for 4 weeks may be considered in patients with a shorter duration of symptoms.

In effect, it is a mistake. Patients who were treated for 4 weeks had a shorter duration of symptoms before admission. As noted in previous research, it has traditionally been considered that treatment of this infection should be at least 6 weeks in patients with longer duration of symptoms or with perivalvular abscess 

4. Does severity of symptoms correlate with a worse outcome? For example, perivalvular abscesses, vegetation size, valve perforation, ...

A sentence has been added about the absence of influence on the prognosis (infection recurrence and mortality) of echocardiography findings as the presence of perivalvular abscess, valve perforation or a vegetation size larger than 10 mm

5. Data represented in table 1 will be more clear if this table is divided in to two tables: one with the baseline characteristics and another one with the data on outcome.

Table 1 has been broken down into 2 tables on the advice of the reviewer

---

## [Editor Report · Decision Letter 1]

20 Jul 2020

Four weeks versus six weeks of ampicillin plus ceftriaxone in Enterococcus faecalis native valve endocarditis: A prospective cohort study.

PONE-D-20-05909R1

Dear Dr. Ramos,

We’re pleased to inform you that your manuscript has been judged scientifically suitable for publication and will be formally accepted for publication once it meets all outstanding technical requirements.

Kind regards,

Cécile Oury

Academic Editor

PLOS ONE
---

## [Editor Report · Acceptance letter]

23 Jul 2020

PONE-D-20-05909R1 

Four weeks versus six weeks of ampicillin plus ceftriaxone in Enterococcus faecalis native valve endocarditis: A prospective cohort study. 

Dear Dr. Ramos-Martínez:

I'm pleased to inform you that your manuscript has been deemed suitable for publication in PLOS ONE. Congratulations! Your manuscript is now with our production department. 

Kind regards, 

on behalf of

Dr. Cécile Oury 

Academic Editor

PLOS ONE